# New Variants of Pseudomonas *aeruginosa* High-Risk Clone ST233 Associated with an Outbreak in a Mexican Paediatric Hospital

**DOI:** 10.3390/microorganisms10081533

**Published:** 2022-07-29

**Authors:** Pamela Aguilar-Rodea, Elia L. Estrada-Javier, Verónica Jiménez-Rojas, Uriel Gomez-Ramirez, Carolina G. Nolasco-Romero, Gerardo E. Rodea, Benjamín Antonio Rodríguez-Espino, Sandra Mendoza-Elizalde, Cesar Arellano, Beatriz López-Marcelino, Daniela de la Rosa Zamboni, Ana Estela Gamiño-Arroyo, Rosalia Mora-Suárez, Margarita Torres García, Isabel Franco Hernández, Israel Parra-Ortega, Guillermina Campos-Valdez, Norma Velázquez-Guadarrama, Irma Rosas-Pérez

**Affiliations:** 1Área de Genética Bacteriana, Unidad de Investigación en Enfermedades Infecciosas, Hospital Infantil de México Federico Gómez, Mexico City 06720, Mexico; qbp_pam@hotmail.com (P.A.-R.); darkfire14@live.com.mx (E.L.E.-J.); verozenemij@hotmail.com (V.J.-R.); urielgoramirez93@outlook.es (U.G.-R.); carolinagnolascor@gmail.com (C.G.N.-R.); ge_rodm@hotmail.com (G.E.R.); hipsme@hotmail.com (S.M.-E.); carellanoaipn@yahoo.com.mx (C.A.); bettymar98@gmail.com (B.L.-M.); guillecvg@yahoo.com.mx (G.C.-V.); 2Posgrado en Ciencias de la Tierra, Centro de Ciencias de la Atmósfera, Universidad Nacional Autónoma de México, Mexico City 04510, Mexico; 3Laboratorio de Aerobiología, Centro de Ciencias de la Atmósfera, Universidad Nacional Autónoma de México, Mexico City 04510, Mexico; iarp@atmosfera.unam.mx; 4Programa de Posgrado en Ciencias Químicobiológicas, Escuela Nacional de Ciencias Biológicas, Instituto Politécnico Nacional, Mexico City 11340, Mexico; 5Laboratorio de Investigación y Diagnóstico en Nefrología y Metabolismo Mineral Óseo, Hospital Infantil de México Federico Gómez, Mexico City 06720, Mexico; rodriguezespinoba@gmail.com; 6Subdirección de Atención Integral al Paciente, Hospital Infantil de México Federico Gómez, Mexico City 06720, Mexico; rzdaniela@hotmail.com; 7Departamento de Epidemiología, Hospital Infantil de México Federico Gómez, Mexico City 06720, Mexico; analetse@hotmail.com (A.E.G.-A.); rosymora66@live.com.mx (R.M.-S.); matoga_17@yahoo.com.mx (M.T.G.); 8Departamento de Laboratorio Clínico, Hospital Infantil de México Federico Gómez, Mexico City 06720, Mexico; isafrah1967@gmail.com (I.F.H.); i_parra29@hotmail.com (I.P.-O.)

**Keywords:** *Pseudomonas aeruginosa*, multidrug resistance, high-risk clones, ST233, ST235, outbreak

## Abstract

Recent multidrug resistance in *Pseudomonas aeruginosa* has favoured the adaptation and dissemination of worldwide high-risk strains. In June 2018, 15 *P. aeruginosa* strains isolated from patients and a contaminated multi-dose meropenem vial were characterized to assess their association to an outbreak in a Mexican paediatric hospital. The strains were characterized by antibiotic susceptibility profiling, virulence factors’ production, and biofilm formation. The clonal relationship among isolates was determined with pulse-field gel electrophoresis (PFGE) and multi-locus sequence typing (MLST) sequencing. Repressor genes for the MexAB-OprM efflux pump were sequenced for haplotype identification. Of the strains, 60% were profiled as extensively drug-resistant (XDR), 33% as multidrug-resistant (MDR), and 6.6% were classified as sensitive (S). All strains presented intermediate resistance to colistin, and 80% were sensitive to aztreonam. Pyoverdine was the most produced virulence factor. The PFGE technique was performed for the identification of the outbreak, revealing eight strains with the same electrophoretic pattern. ST235 and ten new sequence types (STs) were identified, all closely related to ST233. ST3241 predominated in 26.66% of the strains. Twenty-five synonymous and seventeen nonsynonymous substitutions were identified in the regulatory genes of the MexAB-OprM efflux pump, and *nalC* was the most variable gene. Six different haplotypes were identified. Strains from the outbreak were metallo-β-lactamases and phylogenetically related to the high-risk clone ST233.

## 1. Introduction

*Pseudomonas aeruginosa* is a free-living microorganism that can survive and grow in a nutrient limitation microenvironment [1]. This bacterium is considered as one of the main opportunist pathogens in hospital wards, capable of contaminating a wide variety of objects, solutions, medicines, and even disinfectants, which can all lead to infection in immunocompromised patients [2]. *P. aeruginosa* stands out worldwide as one of the main causes of an important rate of healthcare-associated infections (HAIs), defined as infections acquired within a healthcare setting during patients’ stays and not present during hospital admission [3,4]. 

A multicentre study carried out in Mexico showed that *P. aeruginosa* causes 24% of HAIs in paediatric patients, showing special resistance to carbapenems, quinolones, and third generation cephalosporins, with a major frequency in intensive care units (ICUs) and highly involved in outbreaks [4].

Epidemic HAIs usually occur during outbreaks [5], defined as an unusual increase in the number of infections (two or more) related to each other, and caused by the same pathogen with geotemporal associations [6]. When researching suspected outbreaks, a list of potential patients and several hospital environmental factors must be considered, such as the clinical records of the patients, including treatments, devices, or material used in their care, geographic location of hospital wards, possible interactions among patients, common cleaning supplies, common health personnel, and the moment where the disease is taking place, among others [6,7]. 

Although epidemic outbreaks are infrequent, they represent an important problem due to the increase in morbidity and mortality rates [6]. The genetic relationship of the suspected strains associated within an outbreak must be confirmed or refuted, to determine the source of infection, possible reservoirs, routes of transmission, and therefore control their spread [6,8]. Next-generation whole genome sequencing (WGS) has become the new gold standard for bacterial typing due to its accuracy for not only epidemiological research in hospital environments, but also different sources, and its capability to generate highly supported phylogenetic trees. However, clustering of epidemiologically linked isolates obtained by WGS is difficult to set due to the typing method and the bacterium, additional to sharp discrimination of clonal populations and sequencing costs. Pulsed-field gel electrophoresis (PFGE) has shown to be as accurate as WGS-based typing by revealing to be less affected than WGS-based typing by accelerated genetic drift, which usually occurs in epidemic *P. aeruginosa*, demonstrating that the technique can still be employed for identification of outbreak-associated strains, additional to the multi-locus sequence typing (MLST) of the *P. aeruginosa* core genome genes [9]. 

In recent decades, multidrug-resistant *P. aeruginosa* high-risk clones have been frequently implicated in hospital outbreaks [7], being responsible for the increasing rates of morbidity and mortality worldwide [10]. High-risk clones exhibit a great capability to accumulate mutations, resistance genes (extended-spectrum β-lactamases or carbapenemases-encoding genes), and exotoxins, allowing their intrahospital persistence, transmission, and their association within multidrug-resistance dissemination among bacterial species [11,12,13,14]. Worldwide, a group of successful high-risk clones (ST111, ST175, ST235, ST233, and ST253) stands out due to the characteristics mentioned above [15,16,17,18], additionally to multidrug resistance, high biofilm production, low pyocyanin and pyoverdine production, and motility, possibly due to the physiological cost that the expression of resistance determinants produces to the bacteria [10,19]. In Mexico, our group first reported the ST233 and ST1725 clones, both registered in the periods of 2007 and 2013, which stand out as the first identification of an extensively drug resistant ST233 clone [18]. 

The identification and notification of main sequence types (STs) involved in outbreaks in different parts of the world apport relevant information to their spread control, to determine the sources of infection, reservoirs, and possible routes of transmission [8]. In addition, it has been observed that identical STs, or phylogenetically related STs, share similar characteristics of multidrug resistance, virulence factors, and haplotypes (mutational patterns) in the repressor genes of the constitutive MexAB-OprM efflux pump (*mexR, nalC,* and *nalD*) [20] characteristics that could be used for their control and eradication.

According to the history of circulation of multidrug-resistant *P. aeruginosa* clones in a paediatric hospital, the aim of this study was to characterize the *P. aeruginosa* isolates associated with an outbreak, in different wards of a third-level healthcare institute in Mexico in June 2018, to establish the source and identify the clone or clones that caused this event.

## 2. Materials and Methods

### 2.1. Biological Samples

In this study, a total of 15 isolates, which were associated with an outbreak from the Hospital Infantil de México Federico Gómez (HIMFG) reported in June 2018, were analysed: 14 strains were isolated from 10 paediatric patients, and 1 from a multi-dose meropenem vial. The patients were in different and distant ward rooms: emergency room, surgical therapy (STx), neonatal intensive care unit (NICU), oncology, and infectious diseases outpatient. Patients who were associated to the outbreak were found to be geographically distant from each other. Identification of the isolates was preliminarily carried out by culturing in both selective and differential media (Mueller-Hinton, Cetrimide (BD BBL, Sparks, MD, USA)), macroscopic and microscopic morphology, and traditional biochemical tests. Confirmation of bacterial identity was performed with the MALDI-TOF automated system (Biomerieux, Marcy l‘Etoile, France). 

### 2.2. Susceptibility Profile 

The susceptibility profiles to 14 antibiotics from 9 different categories were evaluated by the minimal inhibitory concentration method (MIC), through the microdilution technique, according to the manual of the Clinical Laboratory Standard Institute (CLSI) [21]. Fosfomycin was evaluated by the plate dilution technique. The antibiotics tested in this study were: Gentamicin (GEN), Tobramycin (TOB), Amikacin (AK), Meropenem (MEM), Imipenem (IMI), Ceftazidime (CAZ), Cefepime (CPM), Ciprofloxacin (CIP), Levofloxacin (LEV), Carbenicillin (CB), Piperacillin/Tazobactam (P/T), Aztreonam (AZT), Fosfomycin (FOS), and Colistin (CS). All antibiotics employed were from Sigma-Aldrich, St Louis, MO, USA. The *Pseudomonas aeruginosa* ATCC 27853 and *Escherichia coli* ATCC 25922 (American Type Culture Collection, Manassas, VA, USA) reference strains were both employed for validation of all techniques. The breakpoint values (µg/mL) employed for the interpretation of the MIC for the *P. aeruginosa* clinical strains were according to the CLSI, 2021 [21]. 

Once the susceptibility profiles were obtained, all strains were classified as sensitive (S), intermediate (I), resistant (R), multidrug-resistant (MDR), extensively drug-resistant (XDR), or pan drug-resistant (PDR), as established by Magiorakos et al. [22]. 

### 2.3. Phenotypic Screening and Detection of Carbapenemases

Detection and typing of carbapenemases (serine carbapenemase or metallo-β-lactamase) was performed using the modified carbapenems inactivation method (mCIM) and the EDTA-modified carbapenem inactivation method (eCIM), both described by the CLSI, 2021 [21].

### 2.4. Phenotypic Production of Pyocyanin and Pyoverdine

Phenotypic production of pyocyanin and pyoverdine was determined by measuring the absorbance of both pigments at 520 and 407 nm, respectively, in an Epoch Microplate Spectrophotometer (BioTek, software Gen5TM, Winooski, VT, USA). Briefly, *P. aeruginosa* strains were grown on blood agar at 37 °C for 24 h. Colonies were recovered and cultured in Mueller-Hinton broth and adjusted to 1 × 10^6^ colony forming units per millilitre (CFU/mL) (with and without antibiotic), placing 1 mL in a 24-well microplate, and incubated at 37 °C for 24 h. Before incubation, absorbance measuring at 520 and 407 nm (initial absorbance) was performed. Then, 800 μL of these cultures and 480 μL of chloroform (Sigma-Aldrich) were both added to a tube and homogenized, and the organic phase (pyoverdine) was measured at 407 mn (final absorbance). From the same tube, 300 μL of the aqueous phase was placed in a new tube with 800 μL of 0.2 N HCl (Sigma-Aldrich) and homogenized. Finally, the absorbance of the aqueous phase at 520 nm (pyocyanin) was measured (final absorbance). Three replicates were performed per strain. The initial absorbance was subtracted from the final absorbance.

The concentration of antibiotic tested corresponded to the dilution prior to the MIC value determined for each antibiotic, for each of the evaluated strains. A heatmap diagram was built with the obtained results, using the RStudio v. 0.01 program (Vienna, Austria) [23]. The concentration of pyocyanin was calculated in µg/mL, by multiplying the obtained absorbance by the factor 17.072 [24,25]. 

### 2.5. Phenotypic Production of Biofilm

Phenotypic production of biofilm was determined by the crystal violet staining method [26]. Briefly, *P. aeruginosa* strains were grown in TSA broth (BD BBL) for 24 h at 37 °C. A 1:100 dilution was performed, to place 100 μL per well in a 96-well microplate, previously filled with 100 μL of TSA per well (with and without antibiotic). Absorbance at 550 nm (initial absorbance) was measured before incubation at 37 °C for 24 h. Then, the microplates were washed with 200 μL of PBS, 1X per well. The supernatant was removed and fixed with heat for a second wash, as described above. The microplate was then stained with 200 μL of 0.5% crystal violet for 10 min, and two washes were performed with PBS 1X. Finally, 200 μL of alcohol–acetone 1:1 was added to quantify the absorbance at 550 nm. Eight replicates were performed per strain. The initial absorbance was subtracted from the final absorbance. The concentration of antibiotic tested was the dilution prior to the MIC value for each antibiotic, for each evaluated strain. A heatmap diagram was built with the obtained results, using the RStudio v. 0.01 program (Vienna, Austria) [23]. 

### 2.6. Pulsed-Field Gel Electrophoresis (PFGE)

All *P. aeruginosa* strains were cultured on blood agar for 18 h at 37 °C. The isolates were treated with lysozyme and proteinase K (Sigma-Aldrich), embedded in 1.2% agarose blocks, and stored in cell lysis buffer (Tris 1 M pH = 8.0, EDTA 0.5 M pH = 8.0, Sarcosil 10%) and proteinase K overnight. Enzymatic digestion was performed with the *SpeI* enzyme (Jena Bioscience, Jena, Germany), at 37 °C for 2.5 h. Electrophoretic running was performed by pulsed-field in a high-fidelity 1.2% agarose gel, with the CHEF-DRII equipment (Bio-Rad Life Science Research, Hercules, CA, USA), described as follows: an initial pulse of 5 s, final pulse of 40 s, 5.2 V/cm^2^, and maintaining a constant temperature of 14 °C for 22 h. After ethidium bromide (BrEt) (Sigma-Aldrich) staining and washing, the gel was visualized on an iBright CL1000 photo documenter (Thermo Fischer Scientific Inc., Waltham, MA, USA).

For the analysis of the electrophoretic patterns, a binary matrix (presence–absence of bands) was created. A phylogenetic tree was built applying the unweighted pair group method with arithmetic mean (UPGMA) in RStudio v.0.01 (Vienna, Austria) [23], a similarity ≥90% was established as the breakpoint value to consider the clusters reliable. In addition, according to criteria of Tenover et al. [27], based on the differences in the number of bands, the electrophoretic patterns were categorized as indistinguishable, closely related, possibly related, or unrelated to the outbreak.

### 2.7. Multi-Locus Sequence Typing (MLST)

Isolation of bacterial genomic deoxyribonucleic acid (DNA) was performed using the Wizard Genomic DNA purification kit (Promega, Madison, WI, USA) following the manufacturer’s instructions. For the multi-locus sequence typing (MLST) technique, the amplification and sequencing of seven housekeeping genes (*acsA, aroE, guaA, mutL, nuoD, ppsA,* and *trpE*) were both performed with primers and conditions described by Curran et al. [28]. Amplicon sequencing was performed with the BigDye Terminator v3.1 Cycle Sequencing Kit (Thermo Fischer Scientific Inc., Waltham, MA, USA) followed by the BigDye Xterminator Purification Kit (ThermoFischer Scientific Inc., Waltham, MA, USA), and using an ABI 310 Genetic Analyser (Applied Biosystems, Foster City, CA, USA). The resulting electropherograms were manually analysed with the FinchTV 1.5 (Geospiza, Inc.; Seattle, WA, USA; http://www.geospiza.com) [29], ClustalX 2.1 (Conway Institute, UCD, Ireland) [30], and Seaview 3.2 (Institut Français de Bioinformatique, France) [31] programs. 

The consensus sequences obtained for each gene of each strain were individually typed in the PubMLST database (http://pubmlst.org/paeruginosa/, Warwick, UK; accessed on 18 May 2022) for the assignment of an allelic number based on worldwide reports. Once the allelic profile was obtained, the ST of each strain was determined. New alleles and STs were added to the global *P. aeruginosa* PubMLST database (accessed on 18 May 2022). 

### 2.8. MexAB-OprM Efflux Pump Repressor Genes’ Characterization

The amplification and subsequent sequencing of the MexAB-OprM efflux pump repressor genes *mexR, nalC,* and *nalD* were performed with the primers and conditions reported by Aguilar-Rodea et al. [20] for the identification of substitutions and determination of haplotypes in all strains of *P. aeruginosa*. 

### 2.9. Phylogenetic Analysis

The genetic relationship among *P. aeruginosa* strains and its possible association with an outbreak was determined by building a phylogenetic network based on the ST of the strains with the eBURST v3 algorithm (available in the global *P. aeruginosa* PubMLST database, Warwick, UK) [32].

## 3. Results

### 3.1. Isolation, Identification, and Characterization of Pseudomonas aeruginosa from Biological Samples

A total of 15 isolates possibly associated with an outbreak in a Mexican paediatric hospital during the month of June 2018 were recovered: 14 from 10 paediatric patients, and 1 from a multi-dose meropenem vial. All strains were identified as *P*. *aeruginosa*. The strains were isolated from patients of different hospital wards, including emergency (21.43%, *n* = 3), neonatal intensive care unit (NICU) (21.43%, *n* = 3), infectious diseases outpatient (14.29%, *n* = 2), oncology (28.57%, *n* = 4), and surgical therapy (STx) (14.29%, *n* = 2). Up to 71.43% of the strains (*n* = 10) were isolated from blood, while 28.57% were recovered from urine (*n* = 4) (Table 1). More than one morphotype was identified in three patients: Patient 1 (P1), strains 4 and 13 with three days apart, Patient 2 (P2), strains 11, 10, and 3 identified in the same blood culture, and Patient 10 (P10), strains 7 and 6 with 14 days apart (Table 1, Figure 1, and Appendix A). The clinical features of the patients are shown in Appendix A. 

According to the timeline (Figure 1), strain 6 was the first culture isolated from patient 10 (P10) in the infectious diseases outpatient room (1 June 2018); then, strains 2 (P5) and 5 (P8) (3 June 2018), which were obtained from emergency and NICU, followed by strain 8 (P7) (6 June 2018) isolated from the NICU. Three days later, strain 1 (P4) was identified in emergency (9 June 2018). Five days later (14 June 2018), strain 9 (P3) was also isolated from emergency. The next day (15 June 2018), strain 7 was isolated from the infectious diseases outpatient room from the same patient (P10) where strain 6 was previously isolated. Strains 3, 10, and 11 were all identified in oncology (16 June 18) and were isolated from the same patient (P2). Around this time, while investigating the cases and the environment, the Department of Epidemiology identified strain 12 in a multi-dose meropenem vial from the NICU (22 June 2018). The next day, strain 13 (P1) was isolated from STx (23 June 2018); two days later, strain 15 (P6) was identified in the NICU (25 June 2018). Strain 4 was isolated from STx (26 June 2018) from the same patient (P1) where strain 13 was isolated as well. Finally, strain 14 (P9) was isolated from oncology (3 July 2018) (Figure 1 and Table 1).

### 3.2. P. aeruginosa Strains’ Susceptibility Profile

Of the strains, 60% (*n* = 9) were classified as XDR, 33% (*n* = 5) as MDR, and 6.6% (*n* = 1) were classified as S. All strains were classified as intermediate to CS, while 80% were susceptible to AZT. In contrast, 80% were classified as resistant to aminoglycosides (GEN and AK) and β-lactams (P/T), and 93.3% as resistant to cephalosporins (CAZ and CPM), carbapenems (IMI and MEM), fluoroquinolones (CIP and LEV), penicillin (CB), and the aminoglycoside TOB (Table 1).

### 3.3. Carbapenemases’ Production

Carbapenemase phenotyping (production of serine carbapenemases or metallo-β-lactamases) was conducted for all analysed strains (Table 1). Two strains produced serine carbapenemases, while twelve strains were positive for metallo-β-lactamase (Table 1). 

### 3.4. Phenotypic Production of Pyocyanin and Pyoverdine

Pyoverdine production by *P. aeruginosa* strains in the absence of antibiotics was determined in an interval of λ_Abs_ = 1.77–2.62 (Figure 2B). The lowest production of pyoverdine was determined in the presence of GEN (strain 3, λ_Abs_ = 0.37), while the highest production was recorded in the absence of antibiotics (strain 5, λ_Abs_ = 2.62) (Figure 2B).

The production of pyocyanin in the absence of antibiotics was determined in an interval of λ_Abs_ = 1.18–1.80 (20.16–30.68 µg/mL) (Figure 2B). The lowest production of pyocyanin was determined in the presence of CS (strain 10, λ_Abs_ = 0.24, 4.11 µg/mL), while the highest production was recorded in the absence of antibiotics (strain 5, λ_Abs_ = 1.80, 30.68 µg/mL). 

Pyoverdine was the virulence factor that showed the highest phenotypic production in the studied strains, with an average λ_Abs_ = 1.47, while pyocyanin showed an average λ_Abs_ = 0.93 (15.88 µg/mL). Up to 73.33% of the strains produced higher amounts of pyoverdine in the absence of antibiotics, 20% showed higher production in the presence of MEM and CPM (strains 12, 11, and 13), and 6.66% of the strains with P/T (strain 14). Additionally, all strains showed a higher production of pyocyanin in the absence of antibiotics (Figure 2B).

### 3.5. Phenotypic Production of Biofilm

The production of biofilm by *P. aeruginosa* strains in the absence of antibiotics was determined in an interval of λ_Abs_ = 1.14–1.77 (Figure 2B). The lowest biofilm production was determined in the presence of MEM and CPM (strain 4, λ_Abs_ = 0.207), and the highest production was observed with CS (strain 9, λ_Abs_ = 2.055) (Figure 2B). On average, the strains showed a biofilm production of λ_Abs_ = 1.00. Up to 40% of the strains produced a greater amount of biofilm in the absence of antibiotics. On the other hand, 40% showed higher biofilm production in the presence of AK, P/T (strains 12, 9, 10, 11, 13, and 14), and AZT (strains 12, 10, 11, 15, 5, and 7), 33.33% with CPM (strains 12, 9, 10, 11, and 13), 26.66% with MEM (strains 12, 9, 10, and 13), 20% with CS (strains 9, 11, and 13), and 13.33% of the strains with CIP (strains 10 and 13) (Figure 2B). 

### 3.6. Pulsed-Field Gel Electrophoresis (PFGE)

Six distinct PFGE patterns were identified among the 15 *P. aeruginosa* isolates (Figure 2A). Eight strains were designated as identical or indistinguishable (1, 2, 3, 9, 10, 11, 13, 15), and formed clade I. All strains isolated from the multi-dose meropenem vial (12) and strain 4 were grouped in clade II. Clade III included strains 5 and 8, while clade IV included strains 6 and 7. Clades III and IV presented a similarity coefficient < 90%, and were designated as unrelated to the outbreak, as well as the strain 14, which belongs to either to clade II or III. The difference between clade I and II was a single band, so the strains were designated as closely related and were considered part of the outbreak, according to the criteria established by Tenover et al. [27] (Figure 2A).

### 3.7. Multi-Locus Sequence Typing (MLST)

Genotypic characterization of *P. aeruginosa* strains by MLST is shown in Table 2. A total of eleven different STs were identified: ST235, ST3237, ST3238, ST3239, ST3240, ST3241, ST3242, ST3749, ST3750, ST3751, and ST3752, and all new STs were uploaded into the worldwide *P. aeruginosa* PubMLST database (Appendix A), except for the previously reported ST235. ST3241 was the most frequent (*n* = 4) among the analysed strains.

### 3.8. Characterization of the MexAB-OprM Efflux Pump Repressor Genes (mexR, nalC, nalD)

A total of 42 nucleotide substitutions, 25 synonymous and 17 nonsynonymous, were identified in the MexAB-OprM efflux pump repressor genes (*mexR, nalC, nalD*) (Table 3). 

In the *mexR* gene, five synonymous and two nonsynonymous substitutions were identified in 26.6% of the strains. In the *nalC* gene, 15 synonymous substitutions were identified in 20% of the strains, and 15 nonsynonymous substitutions: 11 identified in strain 9. The ^212^G→A substitution was identified in 100% of the strains, followed by the ^459^G→T and ^556^G→A mutations, both identified in 60% of the strains. In the *nalD* gene, five synonymous substitutions were identified in 26.6% of the strains. 

A haplotype number was designated to a specific set of mutations in the repressor genes (*mexR, nalC, nalD*) [20]. A total of six different haplotypes were determined. Haplotype 12 was the most frequent, being identified in 60% of the strains (Table 4).

### 3.9. Phylogenetic Analysis

The phylogenetic network built from the identified STs by the eBURST v3 algorithm (PubMLST *Pseudomonas aeruginosa*) [32] and their genetic relationship is described in Figure 3. The STs of the 15 strains analysed in this study, and in addition all the STs identified in the HIMFG by the working group from 2007 to 2015 [18,20] are depicted as well. The identified haplotype (*mexR, nalC, nalD*) in each strain is indicated. Fifty-nine different STs were incorporated (*n* = 106 strains). ST1725 was the most frequent ST (*n* = 34), followed by ST233 (*n* = 6), ST3241 (*n* = 4), ST1724, ST1729, ST1736, ST2559, ST2710, and ST3238 (*n* = 2). The remaining STs were identified in a single strain. Clonal complexes (CC) were defined as the set of STs that descended from the same founding genotype, which in most of the cases corresponded to the high-risk clones reported worldwide. Members of a given CC shared identical alleles (at least five of the seven loci). 

Clade I (ST3237, ST3238, ST3241, ST3242, and ST3749) and clade II (ST3751, isolated from the multi-dose meropenem vial, and ST3238), determined by PFGE, were closely related to the previously identified high-risk clone ST233 (CC233). Most of the STs in CC233 presented haplotype 12 (*mexR-nalC-nalD*) (except for ST3241 and ST3242). Clade III (ST3240 and ST3750, both with haplotype 28) were related to the high-risk clone ST111 (CC111), haplotype 1 [23]. The latter presented variations, additional to those identified in haplotype 28, including *nalC* mutations: A_4_A, S_5_S, A_23_A, I_41_I, R_43_R, G_49_G, E_59_E, S_118_S, Y_137_Y, A_145_A, A_148_A, P_149_P, and S_209_R, and *nalD*: L_99_L. Finally, clade IV (ST3239, haplotype 29, and ST235, haplotype 30) were part of CC235 and closely related to previously identified haplotype 5 STs [23]. The only difference between haplotype 29 and 5 is an additional A_148_A variation in the *nalC* gene (identified in haplotype 5), and between haplotype 30 and 5, an additional variation in the *mexR* gene (L_131_Q) (identified in haplotype 30), and the lack of the A_148_A variation in the *nalC* gene. 

Additionally, a second phylogenetic analysis was performed among different *P. aeruginosa* STs, which included ten previously isolated and reported STs in the HIMFG, and the strains reported in this study (Appendix A). 

## 4. Discussion

Continued use of antibiotics exerts selection pressure for MDR strains, such as *Pseudomonas aeruginosa* high-risk clones. Despite hygienic measures to prevent and control HAIs, dissemination and persistence of MDR strains (mainly high-risk clones) in healthcare institutes remain as the principal causes of outbreaks due to their genetic and phenotypic factors [10,33,35]. 

The HIMFG is a third-level healthcare institute, where almost 40% of patients are admitted for different types of cancer, who usually go through long hospital stays and receive several antibiotic treatments, and this represents a high-risk factor for the HAIs’ establishment, even more so with the hospital history of MDR *P. aeruginosa* strains’ (ST1725 and ST233) dissemination in different wards, prevailing since 2007. These STs were last identified in 2013, significantly increasing the mortality rate up to 17.39% [18]. In June 2018, an outbreak caused by *P. aeruginosa* was identified in the paediatric hospital. A total of 14 *P. aeruginosa* strains from ten paediatric patients, and one strain recovered from a multi-dose meropenem vial, were analysed. 

High microbial resistance rates were observed. Of the strains, 60% were classified as XDR, and 13.33% as MDR, where AZT and CS were the most effective antibiotics against the *P. aeruginosa* strains isolated from the outbreak. Nevertheless, in the HIMFG, AZT-resistant strains have been previously reported [18,20,36]. CS is considered the last treatment option for patients, due to its neuro and nephrotoxic properties [19,35,37]; however, in recent years, it has emerged as the only therapeutic option for PDR infections, showing greater efficacy in combination with other antibiotics [38,39]. Although AZT has been tested to be effective against *P. aeruginosa* clinical strains (86.6% in vitro effectiveness), this sensitivity can be attributed to the lack of genes or plasmids that code for the production of extended-spectrum β-lactamases (ESBLs); in addition, as a consequence of the unavailability of this antibiotic in Mexico, there is no selection of resistant strains containing plasmids, unlike other countries such as Brazil or India, where 40.8% and 41.38% of resistant rates are respectively reported [40,41,42].

Studies from several countries have highlighted the importance of characterization of outbreaks by pathogenic bacteria within healthcare institutes [4,43,44], where several MDR *P. aeruginosa* strains stand out for their association with high dissemination rates [45,46]. Various studies around the world report an increase in *P. aeruginosa* MDR strains that cause HAIs, and these reports range from 28.2% to 69.8% [40,47,48,49]. For this reason, it is not surprising that MDR strains are the cause of outbreaks due to their ability to spread and resist the hospital environment. In our study, 93.3% of the isolates were classified as MDR.

Several virulence factors in *P. aeruginosa*, which are also expressed in unfavourable conditions, play an important role in infectious processes where specific mechanisms for its survival are activated (i.e., pyocyanin and pyoverdine). Nevertheless, in this study, we observed the highest production of both pyocyanin and pyoverdine in the absence of antibiotics, which suggests that the stress generated by antibiotics does not necessarily trigger the production of these virulence factors. On the contrary, their production diminished in the presence of antibiotics. Schalk et al. reported that the presence of gentamycin diminished the production of pyoverdine, probably associated with the inhibition of its transporter proteins [42]. On the other hand, biofilm is produced to help the bacteria to persist in several infectious processes; in our study, most strains diminished biofilm production in the presence of antibiotics. Contrary to this fact, two clones involved in the outbreak increased their production in the presence of antibiotics, representing a high-risk rate in therapy failures. Additionally, AZT improved the production of biofilm in most strains. In fact, Javed et al. [50] reported that strains of *P. aeruginosa* exposed to CS produced higher biofilm rates. Recent studies have demonstrated that MDR strains of *P. aeruginosa* are higher producers of pyocyanin, similar to our study [51,52].

On the other hand, Mullet et al. reported in 2013 [10] that MDR and XDR strains, both high-risk clones, were significantly associated with a reduced production of pyocyanin and a higher production of biofilm. Contrary to this, Horcajada et al. in 2019 [53] suggested that some virulence factors are not related to their susceptibility profile. In our study, we identified the high-risk clone ST235, which interestingly revealed a reduced production of both pigments, as well as biofilm production, maintaining lower levels than those once determined in the reference strain (*P. aeruginosa* ATCC 27853). The results of our study suggest that the employment of antibiotics can affect the production of virulence and resistance factors, such as pyoverdine, pyocyanin, and biofilm.

MDR *P. aeruginosa* strains possess multiple antibiotic-resistant genes, with the efflux pumps as one of the main factors that contribute to *P. aeruginosa* resistance, mainly due to the MexAB-OprM overexpression. This study agrees with several other authors [54,55], and the modulation of the efflux pump by the *mexR, nalC,* and *nalD* regulator genes and its respective mutations, which can affect their function [34,56,57,58]. Some studies have analysed the three repressor genes of the MexAB-OprM efflux pump as a whole set; in one of these, the researchers showed that not all MDR strains overexpress the MexAB-OprM operon, because of the continuous transcription of both *nalC* and *nalD* genes as a consequence of the mutations in these regulators [55]. In this study, however, six different haplotypes (*mexR-nalC-nalD* nucleotide substitutions) were identified: haplotype 12 stands out as previously described in isolates recovered from patients in the HIMFG, during the period of 2007 and 2015 [20], and haplotypes related to the previously identified haplotype 5 that has been associated with fatal outcomes in patients; however, no patient deaths were reported in the present study.

Worldwide, the sequence types ST111, ST175, ST235, ST233, and ST253 stand out, being considered as of special caution for their persistence and ability to spread in multiple environments [15,16,17,18]. In Mexico, the report of the ST233 and ST1725 clones registered between 2007 and 2013 stands out, where the XDR ST233 clone was identified for the first time, showing resistance to CS [18]. In this study, six different electrophoretic patterns were identified by PFGE, and eleven different STs were determined by MLST within the analysed strains. The phylogenetic tree based on the electrophoretic patterns and the phylogenetic network built with the obtained STs showed correlation: both graphics identified a close phylogenetic relationship between nine of the analysed strains and the strain isolated from the multi-dose MEM vial, all considered as variants of the previously identified ST233 and consequently members of the CC233. In addition, ST3240 and ST3750, both closely related strains to the ST111 (CC111), a ST235 strain, and a related ST (ST3239, CC235) are highlighted for being previously identified as high-risk clones as well; nevertheless, they were considered as not related to the outbreak.

It is remarkable that strains from CC235 (strains 6 and 7, ST235 and ST3239, respectively) occurred exclusively in a patient from the infectious disease outpatient ward, 14 days apart, although both strains were phylogenetically closely related, suggesting the association of the use of antibiotics in the success of ST235 over ST3239. Recent whole genome sequencing studies have identified ST235 high-risk clones with novel genetic characteristics, such as the Type IV Secretion Systems, that allows the uptake of foreign genetic material, contributing to the competition among *P. aeruginosa* isolates [59]. In addition, despite that ST235 is considered a high-risk clone, the patient reported no symptoms, which strongly correlates with the low phenotypic expression of virulence factors. 

The NICU is located on the fourth floor, where *P. aeruginosa* isolates conforming CC111 (strains 8 and 5, ST3240 and ST3750, respectively) were recovered and identified. These STs were related to the ST111, but distant from the ST235. The NICU is distantly located from other services by a floodgate, and handwashing with chlorhexidine upon admission is mandatory. We suggest that this geographic isolation from other areas of the hospital explains the independent clonal evolution; however, 16 days apart, an isolate was obtained from a multi-dose MEM vial, and 3 days apart another strain was identified in a patient (P6) (both related to the outbreak). Inquiring, the multi-dose MEM vial was an antibiotic vial shared within the emergency room, where the vial could have been contaminated, explaining the relationship between the isolates of patient 6 (P6, NICU) and patient 1 (P1, STx) with the outbreak. In addition, NICU and emergency patients are frequently visited by surgeons, nurses, paediatricians, or rehabilitation personnel, who also admit STx patients. In fact, strain 15 (P6) was identified in the NICU, after the closely related isolation in STx (strain 13, P1). 

The remaining isolates were first identified in patients from the emergency room, and later in patient 2 (P2) from oncology. We must highlight that P2 was first admitted to emergency before being translated to oncology, and these isolates occurred at intermediate points between those described above and were associated with the ST233 (CC233). According to the bed location in this service (which are very close to each other), we suggest both direct and indirect transmission events between patients because of contamination of surfaces. However, given that CC233 has already been found in different countries and in this hospital was predominantly found in emergency patients, it is difficult to rule out if the strains were imported from other hospitals or were colonizing readmitted patients.

The sequence types ST111, ST233, and ST235 have been reported as high-risk clones with high mortality and spread rates throughout the world in several countries, such as Spain, Brazil, Venezuela, Greece, Russia, Italy, Japan, Sweden, France, and Turkey, among others [10,12,18,60,61,62]. In 2014, our research group reported the isolation and characterization of ST111 from a water source, classifying the isolate as resistant to CB, FOS, and CS [20]. In this study, two variants of this ST (ST3240 and ST3750), isolated from patients and resistant to TOB, IMI, MEM, CAZ, CPM, CIP, LEV, and CB, were identified, suggesting the adaptation of MDR variants to both outside and clinical environments, representing a significant risk due to its possible transmission to patients, constant circulation and prevalence, and the emergence of new, higher resistant and better adapted STs.

There is a record of the ST233 identified in the period of 2007 to 2015 in the HIMFG, exclusively sensitive to AZT [18]. In this study, ten more strains were closely related to this ST, identified as the main cause of the outbreak, which presented an intermediate value of resistance to CS and were classified as sensitive to AZT. Additionally, haplotype 12 (*mexR-nalC-nalD*) (previously reported in ST233) was determined in eight of these strains, suggesting a relationship between the phylogenetically related strains and the haplotype, as previously described [20]. The identification of these variants after the first report of ST233 evidenced the high capability of adaptation and persistence of these clones to the hospital environment, which although they have not yet produced deaths as reported worldwide, they do represent a latent risk in the hospital and may be the cause of future outbreaks [4], HAIs, and resistance reservoirs. In this regard, it should be noted that *P. aeruginosa* infections can be caused by a single ST or by more than one ST belonging to different CCs, hence the importance of determining the STs involved, since they can differ genotypically and phenotypically. Effective treatment to eradicate one ST will not necessarily be functional to eliminate another ST, since virulence and resistance factors can be transferred between bacteria.

In the HIMFG, a permanent hand washing program exists, which was intensified during the outbreak, as well as the continuous hospital wards’ disinfection reinforced with exhaustive washing and UV irradiation, specifically where the contaminated MEM vial was detected. Nursing sessions were implemented to enhance correct disinfection and avoid contamination during intravenous drugs’ preparation, and the process was monitored. The actions taken by the Department of Epidemiology to identify and contain the outbreak prevented fatal outcomes in the patients and successfully controlled the outbreak.

This is the first report in Mexico of the ST235 and its variant ST3239, both classified as XDR, with an intermediate value of resistance to CS and with haplotypes 30 and 29 (*mexR-nalC-nalD*), respectively. Both haplotypes are closely related to previously identified haplotype 5 STs, which were associated with death in paediatric patients, compared to other haplotypes [20].

## 5. Conclusions

Ten out of 15 strains of *P. aeruginosa* were associated with the outbreak by PFGE. Although most strains were identified as new STs, all were closely related to previously identified high-risk clones (ST233) in the HIMFG. This is valuable, especially when high-risk clones identified in the outbreaks are usually considered more important than new variants; however, new variants could be as important as these high-risk clones, by showing similar characteristics, such as the presence of metallo-β-lactamases, the same haplotype of the MexAB-OprM efflux pump, increased production of biofilm, and a decrease in the production of virulence factors (specifically in the presence of antibiotics), favouring better adaptation and persistence mechanisms in the hospital environment. The diversification of high-risk clones is a latent problem, especially in non-optimal environments for this bacterium. For this reason, outbreaks will always be considered as an alarm due to the presence of some of them.

## Figures and Tables

**Figure 1 microorganisms-10-01533-f001:**
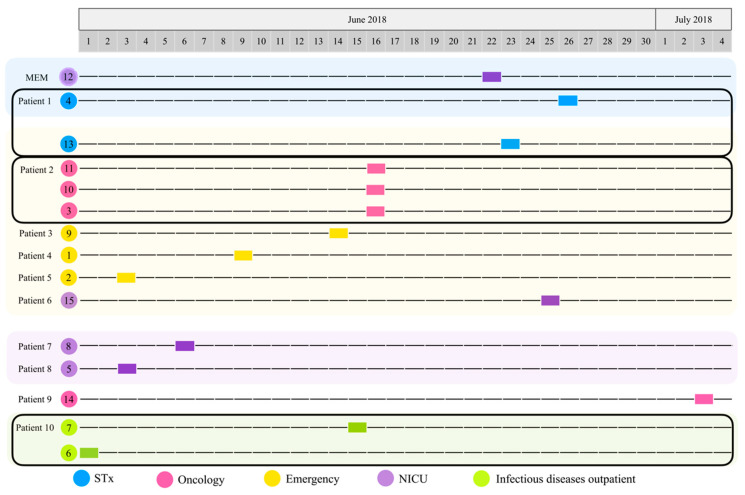
Timeline of the isolation of *P. aeruginosa* strains. Hospital ward: surgical therapy (STx, blue), oncology (pink), emergency (yellow), neonatal intensive care unit (NICU, purple), infectious diseases outpatient (I, green). Patients with two or more isolated strains are highlighted with a black rectangle (Patient 1: strains 4 and 13; Patient 2: strains 11, 10, and 3; Patient 10: strains 7 and 6).

**Figure 2 microorganisms-10-01533-f002:**
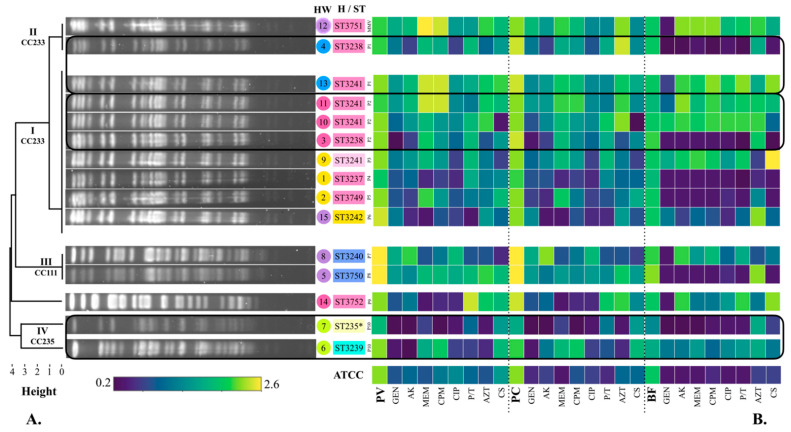
Phylogenetic tree of the 15 *P. aeruginosa* strains based on their electrophoretic pattern (PFGE), phenotypic production of virulence factors, and sequence type (ST). (**A**) Pulsed-field gel electrophoresis (PFGE). Electrophoretic patterns were obtained by PFGE with the *Spe* I enzyme. The phylogenetic tree was built with the unweighted pair group method with arithmetic mean (UPGMA) according to the PFGE electrophoretic patterns. Clades I, II, III, and IV were obtained. Strains from clades I and II were designated as closely related and considered part of the outbreak. CC: Clonal complex. The strain number followed by its identified ST is shown. Colour of the strain is given by the hospital ward (HW) according to Figure 1: Yellow: emergency (E), Blue: surgical therapy (STx), Pink: oncology (O), Purple: neonatal intensive care unit (NICU), Green: infectious diseases outpatient (I). Colour of the ST is given by the *mexR-nalC-nalD* haplotype (H): H12 (pink), H28 (blue), H29 (light blue), H30 (light green), H31 (light pink), H32 (yellow). High-risk clone ST235 is highlighted with a *. Number of patients is shown as well (P1–P10). MMV: Multidose MEM vial. (**B**) Phenotypic production of pyoverdine, pyocyanin, and biofilm. Virulence factors: PV: Pyoverdine, PC: Pyocyanin, BF: Biofilm. Antibiotic categories: 1. Aminoglycosides: GEN: gentamicin, AK: amikacin, 2. Carbapenems: MEM: meropenem, 3. Cephalosporins: CPM: cefepime, 4. Fluoroquinolones: CIP: ciprofloxacin, 5. Penicillins + β-lactamase inhibitors: P/T: piperacillin-tazobactam, 6. Monobactams: AZT: aztreonam, 7. Polymyxins: CS: colistin.

**Figure 3 microorganisms-10-01533-f003:**
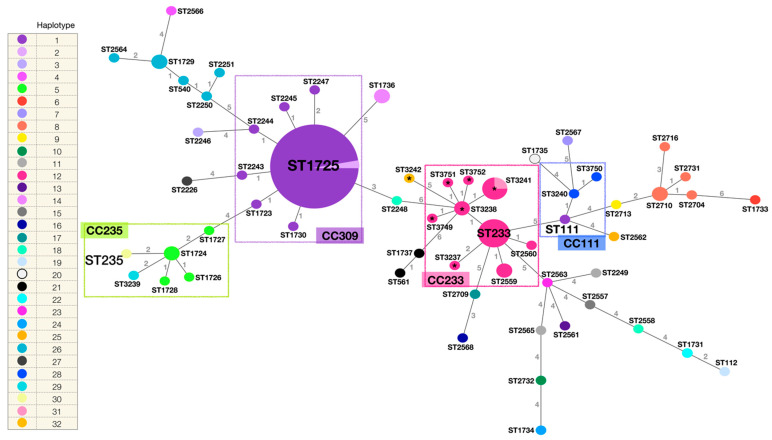
Phylogenetic network of the *P. aeruginosa* STs identified in the HIMFG. STs are indicated. The size of the circumferences is given by the ST frequency, and the numbers in the lines indicate the differences in the allelic profile between the strains (*n* = 7 genes, maximum number of differences). The colour of the ST is given by the identified haplotype (*mexR, nalC, nalD*). Total number of haplotypes = 32. Total number of STs = 59. Total number of strains = 106. Relevant clonal complexes (CC) are highlighted with rectangles (CC235, CC309, CC233, and CC111). Strains related to the outbreak are highlighted with *.

**Table 1 microorganisms-10-01533-t001:** Susceptibility profile of the *P. aeruginosa* strains and their classification.

Strain	Patient	ID	Source	H. Ward	1			2		3		4		5	6	7	8	9	Suscept.	Carb
					GEN	TOB	AK	IMI	MEM	CAZ	CPM	CIP	LEV	CB *	P/T	AZT	FOS *	CS		
**12**	**MMV**	HIM12/18	MMV	NICU	1024	128	256	256	128	32	256	32	32	>1024	512	4	128	2	MDR	M
4	**P1**	HIM4/18	B	STx	512	128	256	256	64	64	128	32	32	>1024	256	8	256	1	XDR	M
13		HIM13/18	B	STx	1024	128	256	256	128	64	256	32	32	>1024	512	4	256	2	XDR	M
11	**P2**	HIM11/18	B	O	1024	128	256	256	128	64	256	4	32	>1024	512	4	256	1	XDR	M
10		HIM10/18	B	O	1024	128	256	256	128	32	256	32	32	>1024	256	4	256	2	XDR	M
3		HIM3B/18	B	O	1024	128	256	256	64	64	256	32	32	>1024	512	8	256	1	XDR	M
9	**P3**	HIM9/18	B	E	1024	128	256	256	128	32	256	32	32	>1024	256	8	256	2	XDR	M
1	**P4**	HIM1/18	B	E	1024	128	512	256	128	32	256	32	16	>1024	256	8	128	2	MDR	M
2	**P5**	HIM2/18	B	E	1024	128	512	256	64	32	256	32	32	>1024	512	8	256	1	XDR	M
15	**P6**	HIM15/18	U	NICU	2	8	8	2	0.25	1	4	0.125	0.5	64	8	4	128	2	S	-
8	**P7**	HIM8/18	U	NICU	0.125	16	16	128	128	32	32	16	16	>1024	64	16	8	2	MDR	M
5	**P8**	HIM5/18	B	NICU	0.125	32	16	128	64	32	32	16	16	>1024	64	8	4	1	MDR	M
14	**P9**	HIM14/18	B	O	1024	128	512	256	128	32	256	32	32	>1024	256	4	128	2	MDR	M
7	**P10**	HIM7/18	U	I	256	128	256	16	128	>1024	512	32	16	>1024	128	64	>1024	1	XDR	S
6		HIM6/18	U	I	256	64	128	32	256	>1024	1024	16	8	>1024	512	256	>1024	1	XDR	S

ID: strain identification number as found in the PubMLST *Pseudomonas aeruginosa* database. Source: blood (B), urine (U), multi-dose meropenem vial (MMV). Hospital ward (H. Ward): emergency (E), surgical therapy (STx), oncology (O), neonatal intensive care unit (NICU), infectious diseases outpatient (I). Antibiotic categories: 1. Aminoglycosides: GEN: gentamicin, TOB: tobramycin, AK: amikacin, 2. Carbapenems: IMI: imipenem, MEM: meropenem, 3. Cephalosporins: CAZ: ceftazidime, CPM: cefepime, 4. Fluoroquinolones: CIP: ciprofloxacin, LEV: levofloxacin, 5. Penicillins: CB: carbenicillin, 6. Penicillins + β-lactamase inhibitors: P/T: piperacillin-tazobactam, 7. Monobactams: AZT: aztreonam, 8. Phosphonic Acid: FOS: Fosfomycin, 9. Polymyxins: CS: colistin. Susceptibility: sensitive (green), intermediate resistant (light yellow), resistant (red). Susceptibility profiles (Suscept.): S: sensitive (green), MDR: multidrug-resistant (yellow), XDR: extensively drug-resistant (orange). Carbapenemases (Carb): S: serine carbapenemase, M: metallo-β-lactamase, -: negative carbapenemase. Colour of the strain number is given by the hospital ward according to Figure 1: Yellow: emergency (E), Blue: surgical therapy (STx), Pink: oncology, Purple: neonatal intensive care unit (NICU), Green: infectious diseases outpatient (I). Breakpoint values for antibiotics highlighted with * are not reported for *P. aeruginosa* in the CLSI 2021 [21].

**Table 2 microorganisms-10-01533-t002:** Allelic profile, sequence types, and haplotypes of the MexAB-OprM efflux pump repressor genes (*mexR, nalC, nalD*) of the analysed *P. aeruginosa* strains.

Strain	*acsA*	*aroE*	*guaA*	*mutL*	*nuoD*	*ppsA*	*trpE*	ST
1	83	5	30	218	4	31	41	3237
2	83	5	30	11	45	31	41	3749
3	16	5	30	11	45	31	41	3238
4	16	5	30	11	45	31	41	3238
5	63	5	5	4	45	4	3	3750
6	82	91	3	13	1	2	4	3239
7	38	11	3	13	1	2	4	235 *
8	17	5	5	4	45	4	3	3240
9	16	5	30	218	45	31	41	3241
10	16	5	30	218	45	31	41	3241
11	16	5	30	218	45	31	41	3241
12	16	5	30	140	45	31	41	3751
13	16	5	30	218	45	31	41	3241
14	16	5	30	216	45	31	41	3752
15	7	5	7	7	45	12	7	3242

* Previously reported ST235. According to data from the worldwide *P. aeruginosa* PubMLST database, last update: 18 May 2022: 405 isolates identified as ST235 are reported worldwide, 24,958 allelic sequences, and 8201 isolates.

**Table 3 microorganisms-10-01533-t003:** Substitutions identified in the *mexR*, *nalC*, and *nalD* repressor genes in *P. aeruginosa* strains.

Repressor Gene	Genetic Variation	Total *n* = 15	Nucleotide Variations	Amino Acid Variation
*mexR*	No substitution	11	-	-
	Synonymous substitution (*n* = 5)	4	60G→A, 264C→T, 327G→A, 384G→A, 411G→A	V20V, **S88S, E109E, Q128Q, Q137Q**
	Nonsynonymous substitution (*n* = 2)	4	377T→A, 392T→A	**V126E**, L131Q
*nalC*	No substitution	0	-	-
	Synonymous substitution (*n* = 15)	3	12T→G, 15T→C, 69T→C, 123A→T, 147G→A, 177G→A, 354C→T, 358C→A, 369G→A, 411T→C, 415C→T, 420C→G, 426G→A, 435C→A, 447T→C	A4A, S5S, A23A, I41I, G49G, E59E, S118S, R120R, A123A, Y137Y, L139L, E142E, A145A, A145A, **P149P**
	Nonsynonymous substitution (*n* = 15)	15	194T→G, 212G→A, 223G→T, 283G→T, 402G→C, 422G→T, 428G→A, 433G→A, 440T→C, 457G→C, 459G→T, 486G→C, 517C→A, 556G→A, 625A→C	V65G, G71E, D75Y, G95C, Q134H, S141I, R143Q, A145T, V147A, E153Q, E153D, Q162H, L173I, A186T, **S209R**
*nalD*	No substitution	11	-	-
	Synonymous substitution (*n* = 5)	4	169C→T, 276C→T, 295T→C, 333C→T, 540C→T	L57L, C92C, L99L, I111I, D180D
	Nonsynonymous substitution (*n* = 0)	0	-	-

Nucleotide variation: number indicates the nucleotide position, first letter indicates reference strain (*P. aeruginosa* PAO1) nucleotide, and the second letter the nucleotide that substitutes the original. Amino acid variation: A: alanine, C: cysteine, D: aspartic acid, E: glutamic acid, F: phenylalanine, G: glycine, H: histidine, I: isoleucine, K: lysine, L: leucine, M: methionine, N: asparagine, P: proline, Q: glutamine, R: arginine, S: serine, T: threonine, V: valine, W: tryptophan, Y: tyrosine. First letter indicates reference strain amino acid, the number indicates the amino acid position, and the second letter indicates the amino acid that substitutes the original amino acid. In bold are genetic variations previously reported by Quale et al. [33] and Suresh et al. [34].

**Table 4 microorganisms-10-01533-t004:** Identified haplotypes of the MexAB-OprM efflux pump repressor genes in *P. aeruginosa* strains.

Strain	ST	Haplotype	*mexR*	*nalC*	*nalD*	Total Mutations
1	3237	**12**		**G_71_E, E_153_D, A_186_T**		**3**
2	3749					
3, 4	3238					
10, 11, 13	3241					
12	3751					
14	3752					
5	3750	**28**	S_88_S, E_109_E, Q_128_Q, Q_137_Q, **V_126_E**	**G_71_E**	L_57_L	7
8	3240					
6	3239	**29**	V_20_V, E_109_E, Q_128_Q, Q_137_Q, **V_126_E**	A_4_A, S_5_S, A_23_A, I_41_I, G_49_G, E_59_E, S_118_S, R_120_R, A_123_A, Y_137_Y, A_145_A, P_149_P, **G_71_E, E_153_Q, S_209_R**	C_92_C, L_99_L, I_111_I, D_180_D	24
7	235	**30**	V_20_V, E_109_E, Q_128_Q, Q_137_Q, **V_126_E, L_131_Q**	A_4_A, S_5_S, A_23_A, I_41_I, G_49_G, E_59_E, S_118_S, R_120_R, A_123_A, Y_137_Y, A_145_A, P_149_P, **G_71_E, E_153_Q, S_209_R**	C_92_C, L_99_L, I_111_I, D_180_D	25
9	3241	**31**		L_139_L, E_142_E, A_145_A, **V_65_G, G_71_E, D_75_Y, G_95_C, Q_134_H, S_141_I, R_143_Q, A_145_T, V_147_A, Q_162_H, L_173_I**		14
15	3242	**32**		**G_71_E**		1
		**Number of haplotypes**	3	4	3	**6**

First letter indicates reference strain (*P. aeruginosa* PAO1) amino acid, the number indicates the amino acid position, and the second letter indicates the amino acid that substitutes the original. Amino acid variation: A: alanine, C: cysteine, D: aspartic acid, E: glutamic acid, F: phenylalanine, G: glycine, H: histidine, I: isoleucine, K: lysine, L: leucine, M: methionine, N: asparagine, P: proline, Q: glutamine, R: arginine, S: serine, T: threonine, V: valine, W: tryptophan, Y: tyrosine. In bold are nonsynonymous substitutions. The number of haplotypes (specific nucleotide substitutions or combined substitutions) by gene is also described. ST: sequence type

## Data Availability

Some of the nucleotide sequences identified in this study were previously reported and are available in the GenBank database under the following accession numbers: *mexR* gene sequences, MT188163 and MT188164; *nalC* gene sequences, MT188183 and MT188186. The new nucleotide sequences obtained in this study were deposited in the GenBank database under the following accession numbers: *mexR* gene sequence, ON015859; *nalC* gene sequences, ON052748 and ON052749; nalD gene sequences, ON052750–ON052752. Accession numbers for the *Pseudomonas aeruginosa* isolates used in this work are available in the public database *Pseudomonas aeruginosa* PubMLST (See Appendix A).

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
