# Peer review of "New Variants of Pseudomonas aeruginosa High-Risk Clone ST233 Associated with an Outbreak in a Mexican Paediatric Hospital"

_microorganisms, 2022, doi:10.3390/microorganisms10081533_

Round 1
Reviewer 1 Report
The manuscript is interesting and well-written. Few minor corrections:
1. In the title, mention the outbreak's location, e.g. at a pediatric hospital in Mexico.
2. Mention the location of the outbreak and the date that occurred in the Abstract and the last paragraph of the Introduction.
3. The border lines in Tables 3 and 4 are truncated and their boldness is inconsistent.
4. In the Conclusions, explain why this study is important for the rest of the world.
Author Response
The manuscript is interesting and well-written. Few minor corrections:
- In the title, mention the outbreak's location, e.g. at a pediatric hospital in Mexico.
Thank you, we have changed the title to: “New variants of Pseudomonas aeruginosa high-risk clone ST233 associated with an outbreak at a pediatric hospital in Mexico”, considering that the universe of P. aeruginosa strains in June 2018 was of 15 strains and among these, 10 strains were associated with the outbreak studied in this paper.
2.Mention the location of the outbreak and the date that occurred in the Abstract and the last paragraph of the Introduction.
Thank you, changes have been made as follows:
Abstract: In June 2018, 15 P. aeruginosa strains isolated from patients and a contaminated multi-dose meropenem vial were characterized to assess their association to an outbreak at a paediatric hospital in Mexico.
Introduction: According to the history of circulation of multidrug resistant P. aeruginosa clones in a paediatric hospital, the aim of this study was to characterize the P. aeruginosa isolates associated with an outbreak, in different wards of a third level health care institute in Mexico in June 2018, to establish the source and identify the clone or clones that caused this event.
- The border lines in Tables 3 and 4 are truncated and their boldness is inconsistent.
Border lines of all the tables have been corrected.
Boldness depends of the table and is described in each footnote:
Table 3: In bold are genetic variations previously reported by Quale et al., 2006 [36] and Suresh et al., 2018 [37].
Table 4: In bold are non-synonymous substitutions
- In the Conclusions, explain why this study is important for the rest of the world.
This is of great value especially if considering that high-risk clones are mainly identified in outbreaks and therefore the focus of attention; however, the diversification of new variants could represent an imminent risk by showing similar characteristics of resistance which mainly include metallo-β- lactamase and the same MexAB-OprM efflux pump haplotype that possibly favor a better adaptation to the environment than those already considered as high-risk clones, with probable relevant consequences.
Reviewer 2 Report
The authors have characterized the P. aeruginosa isolates associated with an outbreak, in a pediatric hospital, to establish the source and identify the clones that caused this outbreak.
I consider that the title should be reformulated to better match the content of the manuscript. I suggest “New variants of Pseudomonas aeruginosa high-risk clone ST233 associated with an outbreak in a pediatric hospital”
At the end of the abstract, a phrase concluding the results of the research should be added.
In the Introduction section, the authors should add a paragraph including general data on the pediatric infections with Pseudomonas aeruginosa.
I think that the last paragraph in the Results Section should be moved to the Discussion Section.
The article of Pérez-Corrales C et al. present important results and should be discussed in Discussion section (doi: 10.1186/s13756-021-00942-7. PMID: 33910633; PMCID: PMC8082860).
„This is the first report in Mexico of the ST235 and its variant ST3239, both classified as XDR, with an intermediate value of resistance to CS and with haplotype 5 and 29 (mexRnalC-nalD), respectively [23].” This paragraph is confusing. It is not clear if the present manuscript or the manuscript doi: 10.1371/journal.pone.0266742 is the first report in Mexico.
The Conclusion section is too long. I consider that some phrases could be moved to the Discussion section.
All manuscript should be revised for typos and to eliminate any other writing errors.
Author Response
Comments and Suggestions for Authors
The authors have characterized the P. aeruginosa isolates associated with an outbreak, in a pediatric hospital, to establish the source and identify the clones that caused this outbreak.
- I consider that the title should be reformulated to better match the content of the manuscript. I suggest “New variants of Pseudomonas aeruginosa high-risk clone ST233 associated with an outbreak in a pediatric hospital”
Thank you, the title has been edited to: “New variants of Pseudomonas aeruginosa high-risk clone ST233 associated with an outbreak in a Mexican paediatric hospital”, considering that the universe of P. aeruginosa strains in June 2018 was of 15 strains and among these, 10 strains were associated with the outbreak studied in this paper.
- At the end of the abstract, a phrase concluding the results of the research should be added.
We have added: “Ten strains from the outbreak were metallo-β-lactamase and phylogenetically related to the high-risk clone ST233.” at the end of the abstract.
- In the Introduction section, the authors should add a paragraph including general data on the pediatric infections with Pseudomonas aeruginosa.
A paragraph in the Introduction Section has been added: “A multicentre study carried out in Mexico showed that P. aeruginosa causes 24% of HAIs in paediatric patients, showing special resistance to carbapenems, quinolones and third-generation cephalosporins with a major frequency in ICU (Intensive Care Unit) and highly involved in outbreaks (Gutierrez Muñoz J., et al., 2017)”.
- I think that the last paragraph in the Results Section should be moved to the Discussion Section.
The last paragraph in the Results Section has been modified: “Finally, clade IV (ST3239, haplotype 29; and ST235, haplotype 30), both part of CC235 and closely related to previously identified haplotype 5 STs [23]. The only difference between haplotype 29 and 5 is an additional A148A variation in the nalC gene (identified in haplotype 5); and between haplotype 30 and 5, an additional variation in the mexR gene (L131Q) (identified in haplotype 30), and the lack of the A148A variation in the nalC gene.”
A phrase was added at the last paragraph in the Discussion Section as well: “This is the first report in Mexico of the ST235 and its variant ST3239, both classified as XDR, with an intermediate value of resistance to CS and with haplotype 30 and 29 (mexR-nalC-nalD), respectively; both haplotypes closely related to previously identified haplotype 5 STs which were associated with death in paediatric patients, compared to other haplotypes [23]”.
- The article of Pérez-Corrales C et al. present important results and should be discussed in Discussion section (doi: 10.1186/s13756-021-00942-7. PMID: 33910633; PMCID: PMC8082860).
„This is the first report in Mexico of the ST235 and its variant ST3239, both classified as XDR, with an intermediate value of resistance to CS and with haplotype 5 and 29 (mexRnalC-nalD), respectively [23].” This paragraph is confusing. It is not clear if the present manuscript or the manuscript doi: 10.1371/journal.pone.0266742 is the first report in Mexico.
This statement corresponds to the present manuscript. The article by Pérez-Corrales was carried out in another country, Costa Rica, and is more focused on the type of carbanemase that is mainly associated with resistance to imipenem and meropenem. In our work, the studied strains were characterized by antibiotic susceptibility profiles to fourteen antibiotics, which leads us to classify these ST235 and ST3239 as XDR.
To our knowledge, there are no other reports for the ST235 in our country with the described characteristics.
The Conclusion section is too long. I consider that some phrases could be moved to the Discussion section.
The paragraphs “P. aeruginosa infections can be caused by a single ST or by more than one ST belonging to different CCs, hence the importance of determining the STs involved, since they can differ genotypically and phenotypically; effective treatment to eradicate one ST will not necessarily be functional to eliminate another ST, since virulence and resistance factors can be transferred between bacteria” and “Given that CC233, which already has been found in different Latin American countries and, that in this hospital was predominantly found in emergency patients, it is difficult to rule out if the strains were imported from other hospitals or were colonizing patients who were readmitted for some reason” have been moved to the Discussion Section.
In addition, the discussion and the conclusion were rearranged to adapt to more specific and clear ideas, which prevented long and extensive discussions.
- All manuscript should be revised for typos and to eliminate any other writing errors.
The entire manuscript have been revised and improved.
Reviewer 3 Report
The work presented by Aguilar-Rodea et al. characterizes a series of Pseudomonas aeruginosa isolated in the same period of time in Hospital Infantil de México Federico Gómez (HIMFG), and thus, considered an outbreak. This is in line with previous isolations of this microorganism in the hospital since 2007. Authors have performed and exhaustive work to characterize these isolates: resistance phenotype, virulence determinants (pigments and biofilm production) and to determine the clonality of them (MLST and PFGE).
However, there are some issues of my concern:
· In general, the paper is too long. Introduction and discussion must be shortened, as well as the number of references.
· Introduction, page 2: “Pulsed-field gel electrophoresis (PFGE) and multilocus sequence typing (MLST) are both considered robust methods for the identification of outbreak-associated strains”, should be correctes. In order to determine of outbreak-associated strains, only PFGE is certain, or WGS. MLST allows the comparison between different laboratories, but two strains with same MLST cannot be clones.
· Material and methods: Susceptibility profile - Authors indicate breakpoints to consider isolates as resistant, inmediate or susceptible, but according to what? CLSI? EUCAS? Others?
· Material and methods: Susceptibility profile - It should be interesting if autorhs could also include and classify their isolates according to Mulet et al (2013), MultiS (susceptible to all tested antipseudomonal agents), ModR (non-susceptible to at least one agent in 1 or 2 classes) (plus MDR, XDR or PDR).
· Material and methods: Phenotypic production of Biofilm - Why alcohol-acetone is used to dissolve crystal violet instead of acetic acid? As is usual and as is described in reference 28.
· Material and methods: Pulsed Field Gel Electrophoresis (PFGE) - SpeI should be in italics.
· Material and methods: Pulsed Field Gel Electrophoresis (PFGE) - Authors could use gelJ programme for the analysis of PFGE patterns, because it is a bioinformatic application ready for the same analysis they did: binary matrix and phylogenetic tree applying UPGMA.
· Material and methods: Multilocus Sequence Typing (MLST) - This part of methods must be shortened, too much information is not necessary.
· Results. Table 1. Susceptibility profile of the P. aeruginosa strains and their classification. - Antibiotics in the table foot are in different order than in the table, please rewrite them in the correct order
· Results. Figure 1 Timeline of the isolation of P. aeruginosa strains - This figure should be included after clades and haplotypes results are explained.
· Results Phenotypic production of pyocyanin and pyoverdine. - Could it be possible that the decrease in pigments production in presence of antibiotics is due to lower amount of bacteria? Do you normalize the results of pigment and biofilm production betwwen CFU after the incubation time? This is very necessary, if not, it is not possible to determine if the strains produce less antibiotic or if, simply, there are less bacteria?
· Results. Figure 2. - I would recommend placing phylogenetic tree near to PFGE images, even a new figure with these data would be useful.
· Results. PFGE - It would have been of great interest to include isolates of the different patterns, previously isolated in the hospital, to check if there are the same bacteria.
· Results. Multilocus Sequence Typing (MLST) - This is a key point: it is of concern to obtain different sequence types for isolates with the same PFGE pattern. Please, could authors check this? In addition, if it is, could you give an explanation to this fact?
· Results. Characterization of the MexAB-OprM efflux pump repressor genes (mexR-nalC-nalD) - This characterization seem useless. Authors should explain what is the importance of the haplotypes obtained with this characterization. Furthermore, is necessary to determine and explain which of the substitutions found is related with pumps hyperproduction, and thus with antimicrobial resistance.
· Results. Figure 3. Phylogenetic network of the P. aeruginosa STs identified in the HIMFG - It is not clear which of these STs are from now and which from the previous study.
· Discussion.
This part of the paper is too extensive and must be re-written and shortened.
In the part of percentages of resistance and MDR rates, authors should take into consideration than most of the studies included to compare data, are long studies in which only one isolate by PFGE pattern is considered. In contrast, most of the strains of their study are indistiguisable.
Author Response
Comments and Suggestions for Authors
The work presented by Aguilar-Rodea et al. characterizes a series of Pseudomonas aeruginosa isolated in the same period of time in Hospital Infantil de México Federico Gómez (HIMFG), and thus, considered an outbreak. This is in line with previous isolations of this microorganism in the hospital since 2007. Authors have performed and exhaustive work to characterize these isolates: resistance phenotype, virulence determinants (pigments and biofilm production) and to determine the clonality of them (MLST and PFGE).
However, there are some issues of my concern:
- In general, the paper is too long. Introduction and discussion must be shortened, as well as the number of references.
A: Thank you, we have improved the entire manuscript.
- Introduction, page 2: “Pulsed-field gel electrophoresis (PFGE) and multilocus sequence typing (MLST) are both considered robust methods for the identification of outbreak-associated strains”, should be correctes. In order to determine of outbreak-associated strains, only PFGE is certain, or WGS. MLST allows the comparison between different laboratories, but two strains with same MLST cannot be clones.
A: Thank you, we have changed the idea to:
Pulsed-field gel electrophoresis (PFGE) stands out as the gold standard method for the identification of outbreak-associated strains; however, newer methods have gained relevance for co-genotype the strains, such as multilocus sequence typing (MLST) [9].
- Material and methods: Susceptibility profile -Authors indicate breakpoints to consider isolates as resistant, inmediate or susceptible, but according to what? CLSI? EUCAS? Others?
A: According to the CLSI, 2021. We have already added this information in the material and methods section.
Material and methods: Susceptibility profile -It should be interesting if autorhs could also include and classify their isolates according to Mulet et al (2013), MultiS (susceptible to all tested antipseudomonal agents), ModR (non-susceptible to at least one agent in 1 or 2 classes) (plus MDR, XDR or PDR).
A: In this particular case, we cannot include and classify the isolates according to Mulet et al (2013) as MultiS and ModR. According to Magiorakos et al 2012, 9 strains were classified as XDR, 5 as MDR, and the remaining strain (strain 15) as Sensitive and could not be classified as MultiS nor MoR, as it was sensitive to almost all antibiotics tested for the exception of TOB, FOS and CS, that although are antibiotics of different categories, the value identified for these antibiotics was intermediate resistant, reason why Mulet et al classification could not be included.
- Material and methods: Phenotypic production of Biofilm - Why alcohol-acetone is used to dissolve crystal violet instead of acetic acid? As is usual and as is described in reference 28.
A: We appreciate your observation; however, reports in the literature mention that alcohol at 95% is capable of dissolving crystal violet of Pseudomonas aeruginosa biofilms adhered to the wells. Other reports performed with Staphylococcus spp., and Acinetobacter baumanii highly recommend acetic acid at 33% and alcohol:acetone in a 1:5 ratio. In our experience, we have observed that alcohol:acetone in a 1:1 ratio for P. aeruginosa achieves a perfect dissolution of the dye. We enclose the references here, so not to increase the number of references in the manuscript. Cheng-Hong et al., 2019, doi:10.3390/molecules24101849; Peeters et al., 2007, doi: 10.1016/j.mimet.2007.11.010; Nørskov et al., 2019, doi:10.1016/j.bioflm.2019.100006.
- Material and methods: Pulsed Field Gel Electrophoresis (PFGE) -SpeI should be in italics.
A: Thank you, we have made this correction.
Material and methods: Pulsed Field Gel Electrophoresis (PFGE) -Authors could use gelJ programme for the analysis of PFGE patterns, because it is a bioinformatic application ready for the same analysis they did: binary matrix and phylogenetic tree applying UPGMA.
A: Thanks for the suggestion. In Supplementary Figure 1 we added, we have used this program; however, we notice that to adjust the size of the electrophoretic patterns, the program distorts the original electrophoretic image, which is reflected in an erasure of the initial bands. When analyzing the electrophoretic patterns manually, no distortion of the original electrophoretic image could be seen (Figure 2); we appreciate your suggestion because it is a really useful program.
- Material and methods: Multilocus Sequence Typing (MLST) -This part of methods must be shortened, too much information is not necessary.
A: Thank you, we have improved this part of the manuscript.
Results. Table 1. Susceptibility profile of theP. aeruginosa strains and their classification. - Antibiotics in the table foot are in different order than in the table, please rewrite them in the correct order.
A: Thank you, we have corrected this part of the manuscript.
- Aminoglycosides: GEN: gentamicin, TOB: tobramycin, AK: amikacin, 2. Carbapenems: IMI: imipenem, MEM: meropenem, 3. Cephalosporins: CAZ: ceftazidime, CPM: cefepime, 4. Fluoroquinolones: CIP: ciprofloxacin, LEV: levofloxacin, 5. Penicillins: CB: carbenicillin, 6. Penicillins + β-lactamase inhibitors: P/T: piperacillin-tazobactam, 7. Monobactams: AZT: aztreonam, 8. Phosphonic Acid: FOS: Fosfomycin, 9. Polymyxins: CS: colistin.
- Results. Figure 1Timeline of the isolation of P. aeruginosa strains - This figure should be included after clades and haplotypes results are explained.
A: Thank you, your suggestion has been taken into consideration; however, we consider that this is the appropriate place for Figure 1 since it corresponds to the universe of study and describes the distribution of the strains and the dates of isolation. We consider it relevant to edit Figure 1 in order to remove the clades and clonal complexes classification from the figure, as these concepts are not previously described and are well explained later in the following figures and tables. We appreciate your valuable input.
- Results Phenotypic production of pyocyanin and pyoverdine. -Could it be possible that the decrease in pigments production in presence of antibiotics is due to lower amount of bacteria? Do you normalize the results of pigment and biofilm production between CFU after the incubation time? This is very necessary, if not, it is not possible to determine if the strains produce less antibiotic or if, simply, there are less bacteria?
A: You are right, In fact we normalized and adjust the amount of bacteria which corresponds to 1x106 CFU/ml in each well microplate and that additionally was corroborated by quantifying the absorbance at 520 nm and 407 nm in an Epoch Microplate Spectrophotometer (BioTek, software Gen5TM, Winooski, VT, USA) prior incubation at 37ºC for 24 hours (Initial absorbance). Then, pigments extraction was performed and the absorbance was measured at 520 nm and 407 nm (Final absorbance). The initial absorbance was subtracted from the final absorbance. In addition, three replicates were performed per strain; so it could not be possible that decrease in pigments or biofilm production in presence of antibiotics is due to different amount of bacteria. Description of the methodology can be found in the material and methods section.
Results. Figure 2. - I would recommend placing phylogenetic tree near to PFGE images, even a new figure with these data would be useful.
A: Thank you, we have improved Figure 2.
- Results. PFGE - It would have been of great interest to include isolates of the different patterns, previously isolated in the hospital, to check if there are the same bacteria.
A: Thank you for the comment and suggestion, we have added a new Supplementary Figure:
Figure 1. Phylogenetic tree of 25 P. aeruginosa strains identified in the HIMFG based on their electrophoretic pattern (PFGE). Which included the 15 strains analyzed in this study and 10 strains previously isolated in the hospital. Reference stains were also considered as well.
Results. Multilocus Sequence Typing (MLST) -This is a key point: it is of concern to obtain different sequence types for isolates with the same PFGE pattern. Please, could authors check this? In addition, if it is, could you give an explanation to this fact?
A: Different sequence type for isolates with the same PFGE patterns is possible, indeed previous studies had reported this fact.
Different molecular typing methods show different discriminatory power. MLST detects nucleotide changes within the amplified gene fragment, a limited number of loci, while PFGE examines both nucleotide changes that are in specific restriction sites and changes that involve large insertions or deletions of DNA, differences in other loci (not included in the MLST) can be visualized.
PFGE and MLST may be considered complementary methods, which are appropriate for studies at distinct scales, i.e. local epidemiology versus global population structure, respectively.
MLST defines genetically groups (CC) derived from a founding genotype or common ancestor; members of a given CC shared identical alleles (at least five of the seven loci). PFGE identifies genetic relatedness by constructing a dendrogram; a similarity >90% was established as breakpoint value to consider the clusters reliable. However, bands of the same size and intensity are assumed to be identical in PFGE, but indistinguishable fragments by size or intensity can occur. In this study, there were three CCs (CC233, CC111 and CC235), two of them with two different PFGE types, but each PFGE type was found in only one CC, which means consistency within a CC. Multiple PFGE types within a given ST possibly suggests insertions and deletions of large fragments of DNA are more common than point mutations (Nemoy et al., 2005, doi: 10.1128/JCM.43.4.1776-1781.2005.).
- Results. Characterization of the MexAB-OprM efflux pump repressor genes (mexR-nalC-nalD) -This characterization seem useless. Authors should explain what is the importance of the haplotypes obtained with this characterization. Furthermore, is necessary to determine and explain which of the substitutions found is related with pumps hyperproduction, and thus with antimicrobial resistance.
A: MDR strains of P. aeruginosa possess multiple antibiotic resistance genes, with efflux pumps being attributed as one of the main factors contributing to P. aeruginosa resistance, mainly due to overexpression of MexAB-OprM. Mutations in the mexR, nalC, and nalD repressor genes can affect their function. As it has been described that haplotypes are highly associated with STs, the identification of haplotypes in this study was carried out as a consequence of a previous study by the working group where it was shown that the haplotype is closely related to genetically close clones, regardless of the resistance shown by the strains. Furthermore, some haplotypes have been associated with fatal outcomes in patients, such as haplotype 5; however, no haplotype 5 strains were identified and no patient deaths were reported in this study.
Results. Figure 3. Phylogenetic network of theP. aeruginosa STs identified in the HIMFG - It is not clear which of these STs are from now and which from the previous study.
A: Strains related to the outbreak in the present study are highlighted with an asterisk (most of them corresponded to the CC233). This is mentioned at the footnote of Figure 3.
- Discussion.
This part of the paper is too extensive and must be re-written and shortened.
A: Thank you, we have improved the entire manuscript.
In the part of percentages of resistance and MDR rates, authors should take into consideration than most of the studies included to compare data, are long studies in which only one isolate by PFGE pattern is considered. In contrast, most of the strains of their study are indistiguisable.
A: The studies we included to compare our data were used for the purpose of placing, in general, the frequency of MDR strains around the world, however, we consider the fact that these studies were not conducted with outbreaks, as mentioned: “however, there are no outbreak data in these type of studies. In our study, 93.3% of the isolates were classified as MDR”.
Round 2
Reviewer 2 Report
I read the comments and the new variant of the manuscript. The manuscript was improved.
Author Response
The authors made the minor changes suggested, we appreciate the comments that improved the manuscript
Reviewer 3 Report
All suggestion and chafes have been followed y authors
Author Response

(The authors gave the same response as above.)
